# Functional Outcomes in Head and Neck Cancer Patients

**DOI:** 10.3390/cancers14092135

**Published:** 2022-04-25

**Authors:** Herbert Riechelmann, Daniel Dejaco, Teresa Bernadette Steinbichler, Anna Lettenbichler-Haug, Maria Anegg, Ute Ganswindt, Gabriele Gamerith, David Riedl

**Affiliations:** 1Department of Otorhinolaryngology—Head and Neck Surgery, Medical University of Innsbruck, 6020 Innsbruck, Austria; herbert.riechelmann@i-med.ac.at (H.R.); teresa.steinbichler@i-med.ac.at (T.B.S.); praxis@hno-am-dez.at (A.L.-H.); maria.anegg@student.i-med.ac.at (M.A.); 2Department of Radiation-Oncology, Medical University of Innsbruck, 6020 Innsbruck, Austria; ute.ganswindt@i-med.ac.at; 3Internal Medicine V, Department of Hematology & Oncology, Medical University Innsbruck, 6020 Innsbruck, Austria; gabriele.gamerith@i-med.ac.at; 4Department of Psychiatry, Psychotherapy, Psychosomatics and Medical Psychology, Medical University of Innsbruck, 6020 Innsbruck, Austria; david.riedl@tirol-kliniken.at

**Keywords:** head and neck neoplasms, health status, functional outcomes, surveys and questionnaires, questionnaire design

## Abstract

**Simple Summary:**

With increasing long-term survivorship of head and neck cancer (HNC), the functional outcomes are gaining importance. Recently, a tool for the rapid clinical assessment of the functional status in HNC-patients based on observable clinical criteria, termed “HNC-Functional InTegrity (FIT) Scales, was validated. Here, the functional outcomes of 681 newly diagnosed HNC-patients were reported using the HNC-FIT Scales. A normal/near-normal outcome in all six functional domains was observed in 61% of patients, with individual scores of 79% for food intake, 89% for breathing, 84% for speech, 89% for pain, 92% for mood, and 88% for neck and shoulder mobility. Clinically relevant impairment in at least one functional domain was observed in 30% of patients, and 9% had loss of function in at least one functional domain. Thus, clinically relevant persistent functional deficits in at least one functional domain must be expected in 40% of HNC-patients. The treatment of these functional deficits is an essential task of oncologic follow-up.

**Abstract:**

With the increase in long-term survivorship of head and neck cancer (HNC), the functional outcomes are gaining importance. We reported the functional outcomes of HNC patients using the HNC-Functional InTegrity (FIT) Scales, which is a validated tool for the rapid clinical assessment of functional status based on observable clinical criteria. Patients with newly diagnosed HNC treated at the Medical University of Innsbruck between 2008 and 2020 were consecutively included, and their status in the six functional domains of food-intake, breathing, speech, pain, mood, and neck and shoulder mobility was scored by the treating physician at oncological follow-up visits on a scale from 0 (loss of function) to 4 (full function). HNC-FIT scales were available for 681 HNC patients at a median of 35 months after diagnosis. The response status was complete remission in 79.5%, 18.1% had recurrent or persistent disease, and 2.4% had a second primary HNC. Normal or near-normal scores (3 and 4) were seen in 78.6% for food intake, 88.7% for breathing, 83.7% for speech, 89% for pain, 91.8% for mood, and 87.5% for neck and shoulder mobility. A normal or near-normal outcome in all six functional domains was observed in 61% of patients. Clinically relevant impairment (score 1–2) in at least one functional domain was observed in 30%, and 9% had loss of function (score 0) in at least one functional domain. The main factors associated with poor functional outcome in a multivariable analysis were recurrence or persistent disease, poor general health (ASA III and IV), and higher T stage. Particularly, laryngeal and hypopharyngeal tumors impaired breathing and speech function, and primary radiation therapy or concomitant systemic therapy and radiotherapy worsened food intake. Clinically relevant persistent functional deficits in at least one functional domain must be expected in 40% of the patients with HNC. The treatment of these functional deficits is an essential task of oncologic follow-up.

## 1. Introduction

Recent advances have significantly improved the survival of patients with head and neck cancer (HNC). This has led to more long-term survivors [1,2,3]. Although survival is the most important outcome for HNC patients [4,5,6], other dimensions of treatment outcome such as physical status and functional abilities, psychological status and wellbeing, social interactions, and economic status are becoming increasingly important as a result of this trend [7,8]. These dimensions of outcome are most often measured with quality of life (QoL) instruments, where QoL means the patient’s subjective perception of their state and abilities in these domains [9,10]. Currently, several instruments are available to assess health-related QoL in HNC patients [11,12,13,14,15,16,17,18,19,20]. However, those QoL measurements are subjected to various psychological factors [21,22], response shifts [23,24], and social desirability biases [25,26,27,28,29], which do not necessarily reflect the severity of functional impairment and symptoms [30,31].

In contrast, functional endpoints measure the degree to which patients can perform an activity. Specifically in the head and neck region, functional endpoints include seeing, hearing, smelling, tasting, speech, breathing, eating, and neck and shoulder mobility. The measurements of these functions can be patient-reported, observer-rated, or measured by objective tests such as a barium swallow [10,32]. Unlike QoL measures, functional endpoints strive for objectivity and inter-individual comparability. Patients with the same functional endpoint should also have the same functional scores. Therefore, it makes sense to link functional scores to external criteria that are observable. There are few instruments for the standardized assessment of HNC-related functional endpoints [16,33,34,35]. We recently developed and validated the Head and Neck Functional Integrity Scales (HNC-FIT Scales) for German speaking HNC-patients. The HNC-FIT Scales represent an instrument for a rapid clinician-rated assessment of functional status during routine oncologic follow-up. Functional domains commonly affected in HNC patients are scored. The scores are calibrated using observable external criteria. They are clinically oriented and indicate, for example, whether the patient is dependent on a tracheostomy, feeding tube, opiates, or antidepressants at each follow-up visit [36].

Here, we report the functional outcome of patients with incident HNC treated at the Department of Otorhinolaryngology, Head and Neck Surgery, Medical University of Innsbruck between 2008 and 2020 using the HNC-FIT Scales. The effects of several clinical covariates including age, gender, tumor site, stage, and treatment modality on functional outcome were investigated. Our objective was to describe the outcomes of HNC survivors in frequently affected functional domains and to identify factors associated with favorable functional outcome. 

## 2. Materials and Methods

### 2.1. Study Population

Patients with newly diagnosed carcinoma of the head and neck that were treated at the Department of Otorhinolaryngology, Head and Neck Surgery, Medical University of Innsbruck between 2008 and 2020 were consecutively included. The exclusion criteria were carcinoma of the thyroid, esophagus, eye, or brain or spinal cord, as well as melanoma, sarcoma, lymphoma, other non-carcinomas, carcinomas of head and neck skin, benign neoplasms, intraepithelial neoplasia, dysplasia, inflammatory pseudotumors, and head and neck metastases from distant primary tumor sites. The patient data were prospectively recorded in the clinical tumor registry of the department. Demographic data were recorded at initial diagnosis and included gender, age in years, and the American Society of Anesthesiology (ASA) physical status score as a simple measure of general health status [37]. ASA scores were dichotomized into ASA I/II and ASA III/IV. Further host factors included smoking history (≤10 vs. >10 pack years) [38] and alcohol consumption (daily vs. less than daily). The recorded medical data included tumor site, which was grouped into oral cavity, oropharynx, hypopharynx, larynx, and other sites. The UICC-TNM staging that was valid at the time of the initial diagnosis was used. For clinical T-, N-, and UICC stage, only the first numerical digit with no further subclassification was used. The clinical response status at the last follow-up visit was grouped into complete remission, recurrent, or persistent disease, as well as second primary HNC. The study was conducted according to the principles of the Declaration of Helsinki (59th version, 21 October 2008). The research protocol was approved by the Ethics Committee of the Medical University of Innsbruck (ethics committee number 1182/2019). All patients had given written consent for their data to be used anonymously. 

### 2.2. Treatment

The treatment modality was recommended by an interdisciplinary head and neck tumor board that is in line with National Comprehensive Cancer Network (NCCN) Guidelines. The treatments included upfront surgical resection, radiotherapy (RT), and systemic therapy (ST). The treatment modalities were categorized into surgery only, upfront surgery and postoperative radiotherapy (PORT), upfront surgery and postoperative concomitant ST and RT, primary concomitant ST and RT, and primary RT. The surgical procedures included transoral laser microsurgery, transfacial or transcervical tumor resections, pedicled or free-flap reconstructions, and uni- or bilateral selective or comprehensive neck dissections. RT was applied as a first-line treatment or as PORT in advanced disease if no high-risk factors were present. High risk factors included involved margins or extracapsular lymph node extension [39]. Patients with high risk factors received postoperative ST and RT, as did patients with advanced disease treated without upfront surgery. RT was usually carried out in conventional fractions with 1.8–2.0 Gy daily, five times a week, as three-dimensional conformal radiotherapy or intensity-modulated radiation therapy. The total dose in the area of an untreated primary tumor or in the region of an untreated primary lymph node metastases was 70–72 Gy. In regions associated with a high risk of existing or persisting tumor cells, the dose was 60 to a maximum of 66 Gy, and in areas of physiological anatomical lymphatic drainage, the dose was about 50 Gy. For postoperative patients with high-risk features and for patients with advanced disease who received primary non-surgical treatment, concomitant systemic therapy (ST/RT) was applied [40,41]. Concomitant systemic therapy consisted of either cisplatin, at 100 mg/m^2^ on days 1, 22, and 43, or 25 mg/m^2^ on days 1–4 and 29–32 [42]. Alternatively, mitomycin C, at 10 mg/m^2^ (max. 15 mg total), was used on days 1 and 29, and 5-fluorouracil, at 600 mg/m^2^ (24 h infusion), was used on days 1–5 and 29–33 [43] and was prescribed for patients not suitable for cisplatin treatment. Alternatively, RT was combined with cetuximab with a loading dose of 400 mg/m^2^ once a week before the start of radiotherapy followed by 250 mg/m^2^ weekly for the duration of the RT in frail patients [44]. Depending on their needs, all patients were offered rehabilitation treatment focusing on swallowing and nutrition, speech, physical therapy, psycho-oncological counseling, assistive devices, and comprehensive interdisciplinary programs. 

### 2.3. Head and Neck Functional InTegrity Scales

The HNC-FIT scale is a matrix of six verbal rating scales reflecting the functional domains of food intake, breathing, speech, pain, mood, and neck and shoulder mobility. Only these higher-level functional domains are recorded, e.g., food intake, but not related specific functions and conditions such as chewing, swallowing, salivating, tasting, aspiration, xerostomia, trismus, or dental problems [45]. Each functional domain is divided into five functional levels (Appendix A). These levels are not solely based on the patient’s or examiner’s subjective assessment, such as no, mild, moderate, or severe. Rather, they are anchored to observable external criteria. For example, breathing function is anchored to the external criterion of the need for a tracheotomy, and food-intake function is anchored to dependence on a feeding tube. 

Physicians received detailed instructions on how to complete the HNC-FIT scales (Appendix A Instructions for clinicians). When taking the interim oncologic history during oncologic follow-up, the treating physician interviewed the patient and marked the respective functional levels in this matrix. This structured face-to-face interview lasted a median of 1.2 min and could be entirely integrated into the taking of the interim medical history [36]. Verbal ratings were numerically coded from 0 (loss of function) to 4 (normal function) Low scores reflected disease related functional impairments. These numeric codes were transferred to the clinical tumor registry by a medical documentation assistant after each follow-up visit. For data evaluation, the relative count (percent) of patients at each level of each functional domain was calculated. In the validation study, most of the healthy control subjects examined reported level 4, corresponding to normal function for all functions recorded, with few reporting level 3, which corresponds to a slightly impaired near-normal function [36]. A functional level <3 did not occur for any of the recorded functional domains in healthy controls. Therefore, for the evaluation we divided the data into levels 0–2, which corresponds to a clinically important dysfunction, and level 3 and 4, which correspond to a near-normal or normal function. The functional data from the last oncologic follow-up visit were evaluated in this study.

### 2.4. Data Analysis

Time intervals between the first diagnosis and the last HNC-FIT scale assessment were grouped into <24 months, >=24 months, <60 months, and >=60 months. The absolute and relative patient counts were tabulated for the five levels of the six functional domains (Table 1). 

The effects of the categorical covariates including gender; age group; ASA-Score; smoking history; alcohol consumption; tumor site; T-, N-, and UICC-stage; treatment modality; and p16 positivity on the dichotomized functional outcome (count of patients with integrity scores 0 to 2 vs. integrity scores 3 and 4) of the six functional domains were compared with Chi-square tests (Table 2). 

Factors with *p*-values < 0.2 in the Chi-square tests were included as independent variables in a binary logistic regression model. The dichotomized functional outcome (count of patients with integrity scores 0 to 2 vs. integrity scores 3 and 4) were modeled as the dependent variable. A main effects model with indicator coding for categorical variables with backstep elimination was used. The epsilon criterion to detect collinearity was set to 10^−5^; the model fit was tested with the Hosmer–Lemeshow test. The Hosmer–Lemeshow tests indicated good model fits, and there was no collinearity problem. The response status was analyzed separately because, unlike the factors, it was not known at the time of initial diagnosis and could also be considered in the choice of therapy. The statistical analyses were performed using SPSS 27 (IBM, Armonk, NY, USA).

## 3. Results

### 3.1. Study Population

During the observation period, 1236 patients with incident HNC were treated at our department. Data from at least one HNC-FIT scale assessment were available in 681 patients, which comprises the study population. Of these, 137 (20%) were female. UICC stage I and II disease were observed in 251 patients (36.8%). Of these, 106 patients had stage I or II laryngeal carcinoma. Further demographic and medical data are outlined in Table 2. In the study population, 79.5% were in complete remission, 18.1% had recurrent or persistent disease, and 2.4% suffered from a second head and neck primary at their last follow-up visit. The median time interval between the initial diagnosis and the last follow-up visit in 681 patients was 46 (95%CI 42 to 50) months (46). The last follow-up was less than 24 months after the date of initial diagnosis in 36%, between 24 and 60 months in 38%, and >=60 months in 26%.

### 3.2. Functional Status at Last Follow-Up

At the last follow-up visit, severe limitations with functional integrity scores of 0 to 2 most frequently affected the functional domain of food intake (21.5%), followed by the functional domain of speech (16.3%; Figure 1). Of 681 patients, 14.3% depended on a gastrostomy or feeding tube, 10.2% on a tracheotomy, 10% were not able to communicate by telephone, 6.7% depended on opioids, 8.2% were on antidepressants, and 4.1% were not able to drive a car or comb their hair due to head and neck stiffness (Table 1).

The frequencies of patients with impaired function (integrity scores 0–2) vs. normal or near-normal function (integrity scores 3 and 4) were analyzed with Chi-square tests. The normal or near-normal functional outcome was recorded, e.g., for food intake in 78% of male patients and 80% of female patients (Chi-square *p* > 0.05). The ASA score, tumor site, T-stage, treatment modality, and response status most frequently affected the functional outcome in a univariate analysis (Table 2). The relative frequencies of the normal or near-normal outcomes (integrity scores 3 and 4) of the six functional domains by tumor site, T-stage, and treatment modality are depicted in Figure 2.

### 3.3. Binary Logistic Regression Analyses

Patients with a complete response at the last follow-up visit had a better functional outcome for food intake (OR 2.8; 95%CI 1.5 to 5.3; *p* = 0.001), pain (OR 6.9; 95%CI 3.6 to 13.4; *p* < 0.001) and neck and shoulder mobility (OR 2.6; 95%CI 1.4 to 4.8; *p* = 0.003) when compared with patients with recurrent or persistent disease (see Appendix A). Considering the outcome results for the individual functional domains independent of response status, the following results emerged.

In the univariate analysis of **food intake**, *p*-values of <0.2 were observed for age, ASA score, smoking, drinking, tumor site, T-stage, N-stage, UICC-stage, and treatment modality (Table 2). The odds ratios needed to achieve normal or near-normal food intake were calculated. In this multivariable model, patients with ASA scores of I/II had 2.4 times higher odds to achieve a normal or near-normal outcome for food intake than patients with ASA scores of III/IV (*p* = 0.001). Further significant factors in this multivariable model included T-stage, tumor site, and treatment modality (Table 3).

In the univariate analysis for **breathing** function, *p*-values of <0.2 were observed for gender, ASA score, smoking, tumor site, T-stage, UICC-stage, p16 positivity, and treatment modality (Table 2). In this multivariable model, patients with ASA scores of I/II had a 2.9 times better chance to achieve a normal or near-normal outcome for breathing than patients with ASA scores of III/IV (*p* = 0.001). Further significant factors in this multivariable model included T-stage and tumor site; however, the *p*-value for p16 status missed the 0.05 limit (Table 4).

In the univariate analysis, *p*-values of <0.2 for **speech** function were observed for gender, age, ASA score, smoking, tumor site, T-stage, UICC-stage, p16 positivity, and treatment modality (Table 1). In the logistic regression model, patients with ASA scores of I/II had a 2.7 times better chance to achieve a normal or near-normal outcome for speech than patients with ASA scores of III/IV (*p* = 0.002). Notably, primary concomitant radiotherapy/systemic therapy, which served as the reference treatment modality, resulted in the highest percentage of a normal or near-normal speech function (Table 5).

In the univariate analysis, *p*-values of <0.2 for **pain** were observed for treatment modality, tumor site, T-stage, N-stage, UICC-stage, and ASA score (general health condition). In the backstep logistic regression model, all variables except ASA score (*p* = 0.01) were excluded. The OR to have no or near-normal disease-related pain was 2.9 (95%CI 1.6 to 5.2; *p* < 0.001).

In the univariate analysis, *p*-values of <0.2 for **mood** were observed for gender and ASA score (general health condition). In the logistic regression model, only gender remained in the model (*p* = 0.02). The OR for men to report a normal or near-normal mood was 2.2 (95%CI 1.1 to 4.4). 

In the univariate analysis, *p*-values of <0.2 for **neck and shoulder mobility** were observed for treatment modality, tumor site, T stage, N stage truncated, UICC stage, p16, and ASA score. In the logistic regression model, patients treated only with surgery (reference primary ST/RT; OR 4.2; 95%CI 1.3 to 14.0; *p* = 0.02) had a better chance of reporting normal or near-normal neck and shoulder mobility.

### 3.4. Poorest Functional Otucome

If the functional domain with the poorest outcome would affect the patient the most, the functional score of the domain with the poorest outcome was determined for each patient. A functional score of zero (worst possible outcome) in at least one domain was observed in 62 patients, whereas 185 patients had a normal functional outcome in all six domains (Table 6). Normal or near-normal function (functional integrity) in all six functional domains was observed in 61% of 681 HNC patients. In other words, almost 40% of HNC patients had at least one serious functional impairment.

## 4. Discussion

Survival, health-related quality of life, treatment-related toxicity, and functional outcome are distinct albeit related outcome parameters in patients with head and neck cancer [32]. We examined the outcomes in six essential commonly affected functional domains in HNC patients [36] by using the disease specific HNC-FIT scales, an intentionally minimalistic, validated assessment tool that enables a time-saving assessment of these functions by the clinician using observable external criteria [36]. An additional major advantage of the tool is the standardized and combined assessment, thereby minimizing the risk to under- or overestimate impairments or to overlook essential ones (Table 6). By reporting the percentages of affected patients and dichotomizing the results into normal (scores 1–2) versus impaired (scores 3–4), these results can be more easily compared with other studies and the exact meaning of the functional results becomes directly apparent (Table 1).

The evaluation was performed in 681 survivors out of 1236 HNC patients that were hospitalized and diagnosed with head and neck carcinomas in our department between 2008 and 2020. Even though this cohort can be considered representative for a tertiary oncology referral center in Central Europe, because of their survival, the patients in this study inherently represent a cohort with positive prognostic factors, which was not representative of HNC patients at the time of initial diagnosis (data not shown). The median follow-up time of the survivors was 46 (95%CI 42 to 50) months [46]—so, the majority of patients may be regarded as long-term survivors. All patients were offered a comprehensive rehabilitation program [47] and most patients took advantage of this opportunity. Not surprisingly, patients with a complete response at last follow-up had a significantly better outcome in food intake (*p* = 0.001), pain (*p* < 0.001) and neck and shoulder mobility (*p* = 0.003) than patients with recurrent or persistent disease. Notably, pain was associated with recurrence/persistence in the multivariable logistic regression (OR 6.9; 95%CI 3.6 to 13.4). Given that response status was the only factor recorded at the last follow-up, its effect on functional outcome was considered separately from the other demographic and medical factors recorded at diagnosis. This did not lead to a relevant bias in the effect estimates of the latter factors (see Appendix A).

The influence of demographic and medical factors such as T stage, tumor site, and treatment modality on important domains of functional outcomes were synoptically depicted in star diagrams (Figure 2). Interestingly, general health had an impact on all functional domains studied (Appendix A), possibly reflecting a higher resilience with better general health status. While all functional domains deteriorated with increasing T stage (Figure 2a), the tumor site primarily affected food intake, breathing, and speech, with particularly poor outcomes in hypopharyngeal cancer (Figure 2b). The functional domain most affected by treatment modality was food intake, with better results in patients treated with upfront surgery, whereas better speech outcomes were observed in patients treated with concomitant systemic treatment and radiotherapy (Figure 2c).

In line with previous publications, **food intake** was most frequently affected in HNC survivors [48]. This functional domain covers various functions or symptoms of nutrition (eating and drinking) including swallowing, dysphagia, or trismus [45]. For the food-intake domain, the observable external criteria included dependence on a feeding tube and normality of diet. In line with the results of a previous study [49], 21.4% of patients were either dependent on a PEG or feeding tube or could only eat liquid or soft food. In a logistic regression with backward elimination, the frequency of normal or near-normal food intake depended on the general health condition (ASA score), smoking history, T-stage, tumor site, and treatment modality (Table 3). These factors have already been identified in previous studies [21,50,51,52,53]. Although food intake after primary RT and primary ST/RT was better in this study than in previous reports [54,55], it was still worse than after upfront surgery (Figure 2c), which is also consistent with previous publications [53,56,57,58]. However, this evaluation included many patients who had received three-dimensional conformal radiotherapy and not intensity-modulated radiation therapy, which yields better swallowing outcomes [59,60].

The observable external criteria for **breathing** function included dependence on a tracheotomy and dyspnea in patients without a tracheotomy. Concerning breathing function, the frequency of normal or near-normal breathing significantly depended on the general health condition, T-stage, and tumor site (Table 4), with hypopharyngeal and laryngeal cancers being associated with the worst outcomes for breathing (Figure 2b). This is in line with a previous report [55]. Although not significant, p16 positive patients had a trend for better odds to achieve normal or near-normal breathing [61]. In line with previous publications, treatment modality had a comparatively low impact on breathing [62,63,64].

Phonation and articulation are the key requirements for the understandability of **speech**. Speech intelligibility in telephone conversations served as an external criterion for this functional domain. Speech was the functional domain most affected by tumor stage; however, consistent with previous data, tumor stage had a major impact on several functional domains [10,49,53,65]. Interestingly, T-stage had a substantially higher impact on functional outcome than N-stage or UICC-stage in all functional domains. Poor understandability of speech in patients with laryngeal and hypopharyngeal cancers has been reported previously [21]. When compared with primary ST/RT, patients treated with upfront surgery had markedly poorer odds to achieve normal or near-normal speech intelligibility. This is in line with previous reports [21,55,66,67]. Normal or near-normal speech was particularly rare in patients receiving upfront surgery followed by concomitant ST/RT (Figure 2c).

**Pain** was classified as a body function in the HNC-FIT scale following the International Classification of Functioning, Disability, and Health [68]. The use of pain medication served as the external criterion for the grading of pain. A standardized pain therapy was carried out in cooperation with the Pain Clinic of the Medical University of Innsbruck in accordance with WHO guidelines [69,70]. Interestingly, 89% of HNC patients reported no or almost no pain and 76.4% reported taking no pain medication at all. This low prevalence of pain in HNC-survivors is not consistent with previously reported data. In most pain studies in HNC survivors, chronic pain, often requiring opiates, is a common health problem [71,72,73,74]. The comparatively low consumption of analgesics may be due to regional differences of analgesic consumption [75] and social desirability bias. In logistic regression, only the ASA score (*p* < 0.001) and T stage (*p* = 0.01) had a significant effect on the proportion of patients with no or almost no pain. This is consistent with previous publications [76,77].

The use of antidepressants served as an objective external criterion for depressed **mood**. In the study population, 87.5% reported normal or near-normal mood. This is in line with recent publications in cancer patients [78,79,80], which reported moderate to severe depression in 10–15% of cancer survivors. This compares to an estimated 4.4% of the general population worldwide [81]. However, HNC patients experienced the highest rates of depressive disorder of all oncology patients [82]. Suicide rates among patients with HNC in the USA are more than three times higher than in the general population [83]. In a logistic regression analysis, men had a higher chance to achieve normal or near-normal mood than women (OR 2.2; 95%CI 1.1 to 4.4; *p* = 0.02), which is in line with a previous publication [84]. A higher prevalence of depression was observed when depression scales were used [85,86]. Bamonti and coworkers reported a correlation between pain and depression [87], and this was also observed in this study.

Another function often impaired by HNC treatments is **neck and shoulder mobility**. In the represented cohort, 87.5% of the patients had normal or near-normal mobility with a prevalence of patients treated only with surgery (Figure 2), especially when compared to patients treated with primary ST/RT (OR 4.2; 1.3 to 14.0; *p* = 0.02). Do and co-authors reported, based on the Neck Pain and Disability Scale (NDI), values of more than 20 in patients with spinal accessary nerve injury after head and neck cancer surgery, which corresponds with mildly impaired neck mobility. Range of motion assessments suggest a reduction in head and neck range of motion below 20% in most HNC patients [88,89,90].

Overall, we observed a better functional outcome in HNC survivors than expected based on previous publications on this topic. In many previous studies on functional outcome, patients with advanced tumor stages and/or functionally problematic tumor sites such as the oral cavity, oropharynx, or hypopharynx were selected [91]. In this study group, 251 patients (36.8%) had UICC stage I and II disease and 106 had stage I or II laryngeal carcinoma with a functionally favorable prognosis. Suarez-Cunqueiro and co-authors, for instance, reported normal speech (i.e., absence of problems) in 36.2% and normal swallowing in 24.6% of 851 patients treated with radical surgery for oral and oropharyngeal cancer [92]. Oozer and co-authors used the Performance Status Scale for Head and Neck Cancer Patients in 79 laryngectomies with a mean age of 64 years. Speech was at least understandable most of the time in 62% and normalcy of diet, including full diet with no restrictions, was reported in 76% [93]. In 240 patients with advanced unresectable oropharyngeal and hypopharyngeal carcinoma treated with an aggressive concomitant ST/RT regimen, 25% of patients were dependent on a feeding tube two years after treatment [94]. Of 166 patients, who were disease free five years following the start of treatment in the RTOG9003 trials, 13 (7.8%) were feeding tube dependent [95]. In 181 surgically treated HNC patients, List and coworkers observed a normalcy of diet with a score of >50 in 52% and an understandability of speech of >50 in 77% [14]. 

Two additional factors, which potentially influence the better functional outcome observed in this study, should be discussed. Firstly, the inclusion criteria for the present study were comparatively exclusive. Rather than including all upper aerodigestive tract cancers, only HNC patients typically treated by head and neck surgeons at our institution were included. Consequently, patients suffering from esophageal cancer who typically showed significant functional impairment, for example, were excluded. 

Secondly, approximately one third of the included patients’ tumor sites were categorized as “other site” (see Table 2). Patients subsumed to this group mostly suffered from salivary gland or sinonasal cancer. The better functional outcome observed in this study (Table 2, Table 3, Table 4 and Table 5) might be caused by the design of the HNC-FIT Scales itself. While cancer of the oral cavity, pharynx, and larynx typically impairs functions included by the HNC-FIT Scales, cancer of the salivary glands or paranasal sinuses also leads to functional impairment. However, these impairments might not be identified by the HNC-FIT Scales (e.g., impaired facial movement after radical parotidectomy or impaired nasal breathing after paranasal sinus radiation). The main advantage of the HNC-FIT scales as compact, rapid instruments is also their main limitation. By restricting the scales to the functions and symptoms most frequently mentioned in publications, many important functional domains are ignored [36]. 

One limitation of this study is that not all HNC-related functions and symptoms were explicitly captured by the HNC-FIT scales due to its restriction to higher-level, commonly affected functional domains. In particular, xerostomia, a frequent and often distressing symptom, was mapped into the higher-level function food intake. Other important, but at best only indirectly covered functional outcomes, include aesthetic appearance [96,97], socioeconomic conditions and occupational concerns, fatigue, social eating, sexuality, hearing loss, lymph edema, sleep disturbance, or psychological distress and anxiety [98]. Although the primary aim during the development of the HNC-FIT Scales was to capture the functional outcomes in HNC patients as objectively as possible by anchoring equidistance verbal ratings to objectifiable external criteria [36], certain biases are inherent to the scale (e.g., HNC patients after total laryngectomy will be classified with low functional integrity despite good physical condition). This is the price of being able to capture some sort of functional landscape in many patients with reasonable effort. However, there are abundant data on individual functions, symptoms, or disease-related QoL that take these aspects into account [20,68,99,100].

Another limitation of the present study is that no QoL data were raised. Although the HNC-FIT scales were intentionally designed to objectively assess functional outcomes, the inclusion of subjective single function-assessment tools to assess QoL (e.g., EQ-5D [101]) would have improved the studies quality. However, for previous comparisons of the HNC-FIT scales with the patient-reported EORTC QoL H&N35 questionnaire [18], close correlations were observed [36]. 

Finally, the HNC-FIT Scales were originally developed and validated for German speaking HNC-patients only [36]. Thus, both the generalizability of the observations presented here as well as the applicability of the HNC-FIT Scales for languages other than German are hampered. Although an English translation was provided as part of the original validation study [36], to date no formal translation and cross-cultural validation for English has been performed. 

## 5. Conclusions

The HNC-FIT scales provide an overview of six relevant functional domains in HNSCC patients. The results are consistent with numerous previous studies that have considered individual symptoms or functions separately. According to multivariable analysis, general health status, tumor site, tumor stage, and treatment modality had the strongest impact on functional outcome, with treatment modality being the only selectable factor.

Applying the HNC-FIT scale as a plain tool for the rapid assessment of functional outcomes in HNC patients might aid both patients and physicians. The scale allows for a quick capture and clear presentation of key functional results, filling a gap in HNC outcome assessment. If relevant functional impairments are identified by this simple screening tool, various, more detailed single function assessment tools may supplement the HNC FIT-scales. Thus, an adequate and timely functional rehabilitation may be initiated. 

## Figures and Tables

**Figure 1 cancers-14-02135-f001:**
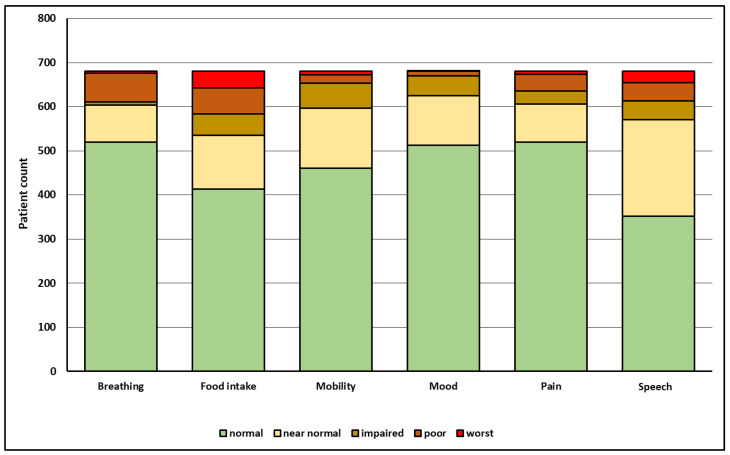
Outcome in six functional domains (x-axis) and respective number of HNC-patients (y-axis) at a median of three years after initial diagnosis as reported by the treating physicians using the HNC-FIT Scale. For related verbal ratings and relative patient count in percent, see Table 1.

**Figure 2 cancers-14-02135-f002:**
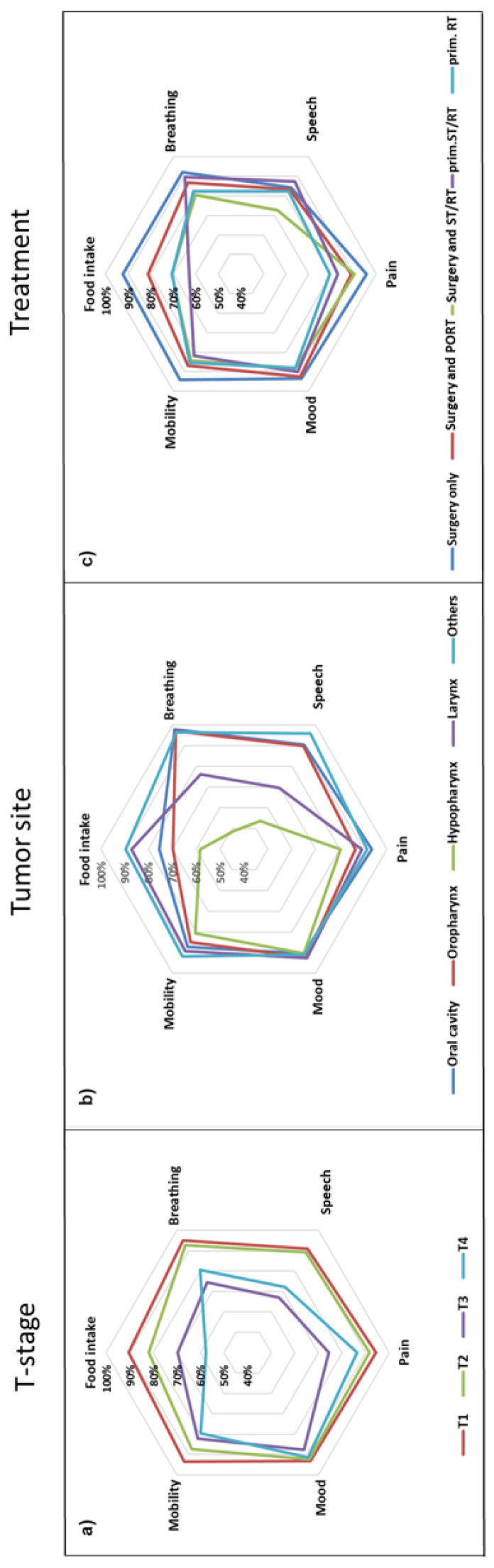
Star plots representing percentage (star axes) of HNC-patients with normal or near-normal functional outcomes (functional integrity) in six functional domains depending on T-stage (**a**), tumor site (**b**), and treatment modality (**c**). The further peripherally, the better (Mobility: neck and shoulder mobility).

**Table 1 cancers-14-02135-t001:** Absolute & relative frequencies (percent) of functional outcomes of 681 HNC-survivors at a median of 4 years after initial diagnosis. The number of patients per functional domain and level as well as the relative proportion in percent are indicated.

Functional Domain	Verbal Rating	Integrity Score	Count	Percent
**Food intake**	No oral feeding; only via gastrostomy tube	0	38	5.6%
	Gastrostomy tube needed; some oral feeding possible	1	59	8.7%
	No gastrostomy tube, oral diet, but only liquid/soft food	2	49	7.2%
	No gastrostomy tube, diet/swallowing near normal	3	122	17.9%
	Normal	4	413	60.6%
**Breathing**	Tracheostoma, needs cuffed cannula	0	5	0.7%
	Tracheostoma, speech cannula/no cannula	1	65	9.5%
	No tracheostoma, breathing difficulties at rest	2	7	1.0%
	No tracheostoma, breathing difficulties only on exertion	3	84	12.3%
	Normal	4	520	76.4%
**Speech**	Not possible, without phonation	0	27	4.0%
	Difficult to understand, no phone calls	1	41	6.0%
	Telephoning possible	2	43	6.3%
	Easy to understand, but pronunciation/voice changed	3	218	32.0%
	Normal	4	352	51.7%
**Pain**	Pain despite opiate therapy	0	7	1.0%
	Controlled with opiates	1	39	5.7%
	Regularly needs non-opioid analgesics	2	29	4.3%
	Needs analgesics from time to time	3	86	12.6%
	Normal	4	520	76.4%
**Mood**	Suicidal thoughts	0	1	0.1%
	Very depressed despite antidepressants	1	10	1.5%
	With antidepressants overall normal mood, very depressed without antidepressants	2	45	6.6%
	Occasionally depressed, no antidepressants needed	3	112	16.4%
	Normal	4	513	75.3%
**Mobility ^1^**	Stiff neck and/or shoulder, hardly any movement possible	0	9	1.3%
	Can hardly comb hair, looking backwards in car not possible	1	19	2.8%
	Combing with problems, looking backwards in car difficult	2	57	8.4%
	Combing and looking backwards in car slightly restricted	3	135	19.8%
	Normal	4	461	67.7%

^1^ Neck and shoulder mobility.

**Table 2 cancers-14-02135-t002:** Relative counts (percent) of 681 patients with incident head and neck cancer with normal or near-normal outcome in HNC functional integrity scales (integrity score 3 and 4) stratified by patients- and disease-related factors.

Factor	Factor-Level	n=	Food	Breathing	Speech	Pain	Mood	Mobility ^1^
**Gender**	male	544	78%	87% * ^2^	81% **	89%	93% *	87%
	female	137	80%	94%	93%	88%	88%	89%
**Age ^3^**	<=50	91	85%	92%	95%*	95%	93%	91%
	51–60	213	81%	88%	85%	88%	93%	87%
	61–70	210	74%	86%	80%	90%	90%	86%
	71–80	129	75%	91%	80%	87%	91%	85%
	>80	38	84%	90%	84%	84%	95%	95%
**ASA ^4^**	ASA I/II	321	82% ***	93% ***	89% ***	93% ***	93%	88%
	ASA III/IV	162	66%	79%	70%	79%	88%	82%
**Smoking**	<10 pack years	219	81% *	91%	85%	89%	91%	87%
	>=10 pack years	266	73%	86%	79%	88%	91%	86%
**Drinking**	<daily	322	78%	89%	82%	89%	90%	86%
	daily	120	69%	87%	78%	88%	93%	83%
**Tumor site**	Lips and oral cavity	94	76% ***	98% ***	90% ***	94%	90%	87%
	Oropharynx	225	70%	97%	90%	87%	92%	85%
	Hypopharynx	41	59%	49%	54%	81%	90%	81%
	Larynx	177	87%	76%	70%	89%	93%	89%
	Others ^5^	144	90%	97%	96%	92%	91%	92%
**T stage**	T0 ^6^	32	81% ***	97% ***	97% ***	91% ***	94%	94% ***
	T1	220	91%	95%	91%	95%	93%	94%
	T2	205	82%	93%	89%	92%	92%	87%
	T3	106	70%	75%	67%	75%	88%	82%
	T4	118	58%	81%	72%	86%	92%	80%
**N stage**	N0	329	88% ***	89%	83%	91% *	92%	91% *
	N1	110	80%	91%	89%	94%	95%	90%
	N2	221	65%	86%	82%	84%	90%	82%
	N3	21	67%	91%	86%	86%	95%	86%
**UICC**	Stage I	163	95% ***	94% *	89%	96% **	94%	95% **
	Stage II	88	81%	91%	86%	90%	90%	86%
	Stage III	127	84%	88%	84%	90%	95%	91%
	Stage IV	303	67%	86%	80%	85%	90%	83%
**p16**	negative	301	74%	85% ***	76% ***	86%	91%	84%
	positive	50	79%	97%	94%	91%	92%	91%
**Observation ^7^**	<=24 months	247	72% ***	84%	80% *	82% ***	90%	88%
	24–60 months	258	87%	93%	89%	93%	92%	89%
	60+ months	176	76%	88%	82%	94%	93%	85%
**Response ^8^**	Complete remission	540	83% ***	92% ***	86% ***	93% ***	92%	90% ***
	Residual disease	123	61%	78%	76%	72%	90%	76%
	Second primary	16	56%	69%	63%	94%	81%	81%
**Treatment ^9^**	Surgery only	251	92% ***	92%	84%	96% ***	94%	94% **
	Upfront surgery & PORT	144	81%	87%	83%	89%	92%	87%
	Upfront surgery & ST/RT	51	71%	80%	73%	90%	88%	84%
	Prim. ST/RT	192	63%	90%	88%	83%	90%	82%
	Prim. RT only	34	71%	82%	82%	79%	88%	85%

^1^ Neck and shoulder mobility; ^2^ The content of this cell means that 87% of 544 male patients had normal or near-normal breathing at last follow-up; ^3^ Age at diagnosis in years; ^4^ American Society of Anesthesiologists score (general health condition); ^5^ This group includes salivary gland cancer and sinonasal cancer; ^6^ Cancer of unknow primary; ^7^ Interval (months) between diagnosis and last functional assessment; ^8^ Response status at last follow-up; ^9^ First line treatment. ST: systematic therapy, RT: radiotherapy, ST/RT: concomitant systemic therapy and radiotherapy; *: *p* < 0.05, **: *p* < 0.005; ***, *p* < 0.0005 (Chi-square test at the respective functional domain).

**Table 3 cancers-14-02135-t003:** Binary logistic model of factors influencing **food intake** in 427 patients with newly diagnosed HNC. Dependent variable was normal or near-normal food intake function (integrity score 3 and 4) vs. impaired food intake (integrity score 0–2). Factors with *p* values < 0.2 in Chi-square tests were included. Odds ratios (OR) indicate the chance to achieve normal or near normal food intake (the higher, the better).

Factor	Factor-Level	B (±SE)	Sig.	OR (95%CI)	Compared to
**ASA ^1^**	ASA I/II	0.82 (±0.27)	0.002	2.3 (1.3 to 3.9)	ASA III/IV
**T stage**	T1	1.43 (±0.42)	<0.001	4.2 (1.8 to 9.5)	T4
	T2	1.06 (±0.34)	0.002	2.9 (1.5 to 5.6)	T4
	T3	0.68 (±0.38)	0.068	2.0 (0.95 to 4.1)	T4
**Tumor site**	Lips and oral cavity	−0.52 (±0.53)	0.328	0.6 (0.2 to 1.7)	Hypopharynx
	Oropharynx	0.06 (±0.45)	0.902	1.1 (0.4 to 2.5)	Hypopharynx
	Larynx	1.05 (±0.52)	0.042	2.9 (1.0 to 7.9)	Hypopharynx
	Others ^2^	1.16 (±0.66)	0.081	3.2 (0.9 to 12.9)	Hypopharynx
**Treatment ^3^**	Surgery only	1.31 (±0.4)	0.001	3.7 (1.7 to 8.1)	Primary ST/RT
	Upfront surgery & PORT	0.32 (±0.33)	0.323	1.4 (0.7 to 2.7)	Primary ST/RT
	Upfront surgery & ST/RT	−0.40 (±0.43)	0.350	0.7 (0.3 to 1.6)	Primary ST/RT
	Prim. RT only	0.34 (±0.62)	0.584	1.4 (0.4 to 4.7)	Primary ST/RT

^1^ Society of Anesthesiologists score (general health condition); ^2^ This group includes salivary gland cancer and sinonasal cancer; ^3^ First line treatment. PORT: postoperative radiotherapy, ST/RT: concomitant systemic therapy and radiotherapy, RT: radiotherapy.

**Table 4 cancers-14-02135-t004:** Binary logistic model of factors influencing **breathing** in 416 patients with newly diagnosed HNC. Dependent variable was normal or near-normal breathing function (integrity score 3 and 4) vs. impaired breathing function (integrity score 0–2). Factors with *p* values < 0.2 in Chi-square tests were included. Odds ratios (OR) indicate the chance to achieve normal or near normal breathing function (the higher, the better).

Factor	Factor-Level	B (±SE)	Sig.	OR (95%CI)	Compared to
**ASA ^1^**	ASA I/II	1.08 (±0.39)	0.006	2.9 (1.3 to 6.3)	ASA III/IV
**T stage**	T1	2.0 (±0.61)	0.001	7.3 (2.2 to 24.6)	T4
	T2	1.37 (±0.52)	0.01	3.9 (1.3 to 11.1)	T4
	T3	0.02 (±0.49)	0.958	1 (0.3 to 2.7)	T4
**Tumor site**	Lips and oral cavity	3.25 (±0.84)	<0.001	25.8 (4.9 to 135.4)	Hypopharynx
	Oropharynx	3.21 (±0.59)	<0.001	24.9 (7.7 to 79.9)	Hypopharynx
	Larynx	1.11 (±0.49)	0.024	3.1 (1.2 to 8.1)	Hypopharynx
	Others ^2^	3.07 (±1.12)	0.006	21.7 (2.4 to 196)	Hypopharynx
**p16 Status**	p16-positive	1.01 (±0.6)	0.092	2.7 (0.8 to 9)	p16-negative

^1^ Society of Anesthesiologists score (general health condition); ^2^ This group includes salivary gland cancer and sinonasal cancer.

**Table 5 cancers-14-02135-t005:** Binary logistic model of factors influencing **speech** function in 416 patients with newly diagnosed HNC. Dependent variable was normal or near-normal speech function (integrity score 3 and 4) vs. impaired speech (integrity score 0–2). Factors with *p* values < 0.2 in Chi-square tests were included. Odds ratios (OR) indicate the chance to achieve normal or near normal speech function (the higher, the better).

Factor	Factor-Level	B (±SE)	Sig.	OR (95%CI)	Compared to
**ASA ^1^**	ASA I/II	1.02 (±0.32)	0.002	2.7 (1.4 to 5.2)	ASA III/IV
**T stage**	T1	2.77 (±0.56)	<0.001	15.9 (5.2 to 48.4)	T4
	T2	2.36 (±0.48)	<0.001	10.6 (4.1 to 27.7)	T4
	T3	0.78 (±0.44)	0.08	2.1 (0.9 to 5.2)	T4
**Tumor site**	Lips and oral cavity	1.94 (±0.63)	0.002	6.9 (2 to 24.2)	Hypopharynx
	Oropharynx	1.85 (±0.49)	<0.001	6.3 (2.3 to 16.8)	Hypopharynx
	Larynx	0.5 (±0.5)	0.314	1.6 (0.6 to 4.4)	Hypopharynx
	Others ^2^	3.2 (±1.15)	0.005	24.5 (2.5 to 233.5)	Hypopharynx
**Treatment ^3^**	Surgery only	−1.92 (±0.5)	<0.001	0.1 (0.05 to 0.3)	Primary ST/RT
	Upfront surgery & PORT	−0.67 (±0.47)	0.152	0.5 (0.2 to 1.2)	Primary ST/RT
	Upfront surgery & ST/RT	−1.93 (±0.59)	0.001	0.1 (0 to 0.4)	Primary ST/RT
	Prim. RT only	−0.58 (±0.7)	0.405	0.5 (0.1 to 2.2)	Primary ST/RT

^1^ Society of Anesthesiologists score (general health condition); ^2^ This group includes salivary gland cancer and sinonasal cancer; ^3^ First line treatment. PORT: postoperative radiotherapy, ST/RT: concomitant systemic therapy and radiotherapy, RT: radiotherapy.

**Table 6 cancers-14-02135-t006:** Poorest outcome in at least one of six functional domains (food intake, breathing, speech, pain, mood, and neck and shoulder mobility) in 681 patients with incident HNC. A normal or near-normal outcome in all six functional domains was observed in 61% of HNC-patients; however, 30% had a relevant impairment in at least one functional domain and 9.1% had maximal functional impairment (worst possible outcome) in at least one functional domain.

Worst Outcome	Frequency	Percent
Worst	62	9.1
Poor	119	17.5
Impaired	85	12.5
Near normal	230	33.8
Normal	185	27.2

## Data Availability

The data presented in this study are available on request from the corresponding author.

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
