# Peer review of "Functional Outcomes in Head and Neck Cancer Patients"

_cancers, 2022, doi:10.3390/cancers14092135_

Round 1
Reviewer 1 Report
This is an interesting study about functional outcomes in 681 head and neck cancer patients.
The paper is well written. However, some issues remain.
Tables 1 and 2 should be moved to the Results section.
The absence of subjective results may represent a limit of the study. Please discuss it.
Author Response
Dear editors, dear referees,
the authors very much appreciate the constructive and positive comments of the referees about the manuscript “Functional outcomes in head and neck cancer patients” (ID cancers-1661528). All suggested corrections and changes are marked with track changes in the main manuscript.
As requested per MDPI we provided an additional Graphical Abstract for the manuscript as well as a blank copy of the patient informed consent form.
The authors would like to thank the referees for their time and effort since their suggested changes substantially improved the quality of the manuscript. Thank you very much for reconsidering our manuscript for publication in Cancers.
Referee 1
- “This is an interesting study about functional outcomes in 681 head and neck cancer patients. The paper is well written.”
All authors involved in this study and the preparation of the manuscript very much appreciate these positive comments.
- “Table 1 and 2 should be moved to the Results section.”
The authors very much appreciate this constructive suggestion. Unfortunately, due to the journals instructions, we were not able to move the tables since Cancer’s Instructions for Authors requires “all figures, schemes and tables should be inserted into the main text close to their first citation and must be numbered following their number of appearance”. All authors involved hope that this does not severely impair the readability of the manuscript.
- “The absence of subjective results may represent a limit of the study. Please discuss it.”
All authors involved in the preparation of the manuscript thank the reviewer for highlighting this important limitation of the study. Consequently, this limitation was included in the Discussion section of the manuscript, as suggested. Please refer to the Discussion section of the manuscript, line 493-498. Moreover, an additional reference was added. Please refer to the References section of the manuscript, line 804-806.

Reviewer 2 Report
- This is a monocentric retrospective study on long term quality of life of patients treated for HN cancers. The authors have performed a huge work to analyse extensively their results. My main concern is that they use a local questionnary, which was self-validated in a previous article, recently published in cancers. Although used in german language, the questionnary is provided in English. A validation should be performed by an English speaking team to make sure that the questionnary is "exportable". This should be specified in the discussion as a limit of the study, and clearly stated in the introduction.
- Inclusion criteriae are questionnable: why did the authors not focus on aerodigestive tract cancers? the "other site" group probably includes salivary gland cancers and sinonasal cancers, subgroups of rare tumours affecting different populations, whose prognosis is far better than the prognosis of SCC, as one can observe in tables 2, 3, 4, 5. This could artificially improve the overall results.
- line 168: <24 months, >=24 months < 60 months, >=60 months
- Table 1: Breathing: blocked cannula: do you mean cuffed cannula?
- A patient having had a total laryngectomy will automatically be classified with the number 1, although his/her quality of life may be better than for patients with difficulties of breathing at rest (number 2).This also implies that the results in table 4 in terms of breathing / tumour site are massively predictible and biased by the scale itself.
This HNC FIT scale appears as a pertinent tool to evaluate QOL and functional symptoms. However, Discussion could have mentioned the fact that this HNC FIT scale was never compared with other QOL Tools, was never used by an other team, was not tested in another language. This may be seen as a problem, as the overall results observed in this study seem quite optimistic when compared with other studies.
The discussion fails to raise the interest of the reader on what could be the practical benefits for the patients and the doctors of adopting the HNC FIT scale in their practice.
- References 5 and 61 are not complete
Author Response
Dear editors, dear referees,
the authors very much appreciate the constructive and positive comments of the referees about the manuscript “Functional outcomes in head and neck cancer patients” (ID cancers-1661528). All suggested corrections and changes are marked with track changes in the main manuscript.
As requested per MDPI we provided an additional Graphical Abstract for the manuscript as well as a blank copy of the patient informed consent form.
The authors would like to thank the referees for their time and effort since their suggested changes substantially improved the quality of the manuscript. Thank you very much for reconsidering our manuscript for publication in Cancers.
Referee 2
2.1. “This is a monocentric retrospective study on long term quality of life of patients treated for HN cancers. The authors have performed a huge work to analyse extensively their results.”
All authors involved in this study and the preparation of the manuscript are very pleased with the honest appreciation about the amount of work involved in the study. Thank you very much.
2.2. “My main concern is that they use a local questionnary, which was self-validated in a previous article, recently published in cancers. Although used in german language, the questionnary is provided in English. A validation should be performed by an English speaking team to make sure that the questionnary is "exportable". This should be specified in the discussion as a limit of the study, and clearly stated in the introduction.”
All authors very much agree on this constructive concern. The HNC-FIT Scales were indeed developed and validated for German speaking HNC-patients recruited in Tirol, Austria. Although an English translation was provided as part of the original validation study, to date no formal translation and cross-cultural adaption of the instrument has been performed. Thus, both, the generalizability of the observations presented in the study as well as the applicability of the HNC-FIT Scales for languages other than German, are hampered. All authors believe this formal translation and cross-cultural adaptation should be performed urgently. As suggested, we aimed at emphasizing this major limitation throughout the manuscript. Please refer to Introduction section, line 70-71 and the Discussion section, line 499-504.
2.3. “Inclusion criteriae are questionnable: why did the authors not focus on aerodigestive tract cancers? the "other site" group probably includes salivary gland cancers and sinonasal cancers, subgroups of rare tumours affecting different populations, whose prognosis is far better than the prognosis of SCC, as one can observe in tables 2, 3, 4, 5. This could artificially improve the overall results.”
The authors very much appreciate these valuable concerns. As defined by the National Cancer Institute the aerodigestive tract comprises of “the combined organs and tissues of the respiratory tract and the upper part of the digestive tract (including the lips, mouth, tongue, nose, throat, vocal cords, and part of the esophagus and windpipe)”. Thus, focusing on aerodigestive tract cancers would include cancer site, which are not typically treated by head and neck surgeons at our institution (e.g. esophageal cancer, tracheobronchial cancer). The inclusion criteria in the present study are representative of the cancer sites typically treated (oral cavity, pharynx, larynx, nose, paranasal sinuses and salivary glands) or and the exclusion criteria are representative of cancer sites not typically treaded (esophagus, thyroid gland, eye, brain, spinal cord, sarcoma, lymphoma) by head and neck surgeons at our institution.
In addition, the authors share the concern of artificially improving the overall results by subsuming salivary gland cancer and sinonasal cancer to the group “other site”. However, the authors believe that this not caused by the better prognosis of these tumor sites but by the HNC-FIT Scales itself: while cancer of the oral cavity, pharynx and larynx typically impairs functions included by the HNC-FIT Scales, cancer of the salivary glands or paranasal sinuses also leads to functional impairment. However, these impairments might not be identified by the HNC-FIT Scales (e.g. impaired facial movement after radical parotidectomy or impaired nasal breathing after paranasal sinus radiation). The authors believe that the main advantage of the HNC-FIT scales as compact, rapid instruments is also their main limitation. By restricting the scales to the functions and symptoms most frequently mentioned in publications, many important functional domains are ignored.
This information has been added as footer accordingly to Table 2, line 182-191, table 3 line 255-262, table 4, line 271-277 and table 5, line 286.293. In addition, both limitation have been added to the Discussion section of the manuscript. Please see the Discussion section, line 459-476.
2.4. “line 168: <24 months, >=24 months < 60 months, >=60 months”
The authors apologize for this imprecision. The mistake was corrected as suggested. Please see the Materials and Methods section, line 169.
2.5. “Table 1: Breathing: blocked cannula: do you mean cuffed cannula?”
The authors apologize for this imprecision. By “blocked cannula” we meant a cuffed cannula indeed. This mistake was corrected accordingly. Please see the Materials and Methods section, Table 1, line 172-175.
2.6. “A patient having had a total laryngectomy will automatically be classified with the number 1, although his/her quality of life may be better than for patients with difficulties of breathing at rest (number 2).This also implies that the results in table 4 in terms of breathing / tumour site are massively predictible and biased by the scale itself.”
The authors very much appreciate this valuable concern. The authors believe that the main advantage of the HNC-FIT scales as compact, rapid instruments is also their main limitation. During the development of the HNC-FIT scales, this issue – amongst others – was raised by the experts involved. However, despite intense discussions and considerable effort no better operationalization for the item breathing could be performed (i.e. improving euqistance between verabl ratings). By anchoring assessments to observable external criteria in order to reduce susceptibility to bias, various other forms of bias are introduced. Although extensively discussed in the previous development and validation study, the authors included an additional paragraph in the Discussion section, to highlight this problem. Please see the Discussion section, line 485-489.
2.7. “This HNC FIT scale appears as a pertinent tool to evaluate QOL and functional symptoms. However, Discussion could have mentioned the fact that this HNC FIT scale was never compared with other QOL Tools, was never used by an other team, was not tested in another language. This may be seen as a problem, as the overall results observed in this study seem quite optimistic when compared with other studies.”
All authors involved in the development and validation of the HNC-FIT Scales highly appreciate this valuable concern.
In terms of a lacking comparison with other QoL tools, the authors would like to highlight the recently published development and validation study. In this previous study, the HNC-FIT Scales were indeed compared to the HNC-specific QoL questionnaire “EORTC H&N35”, to assess criterion validity. The HNC-FIT scale food intake had the highest correlations with the H&N35 subscales ”feeding tube” (r = −0.73, p < 0.001), ”swallowing” (r = −0.72, p < 0.001), and ”social eating” (r = −0.56, p < 0.001). For the HNC-FIT scale “speech”, the highest correlations were found with the H&N35 subscale ”speech” (r = −0.55, p < 0.001), and for the HNC-FIT scale “pain” with the H&N35 subscales ”pain” (r = −0.47, p < 0.001) and ”pain killers” (r = −0.61, p < 0.001). In short, the HNC-FIT scales functional domains correlated closely with the outcome of corresponding scales of the EORTC-HN35-QoL questionnaire, indicating good criterion validity.
In terms of single-center development of the scale in german, all authors agree, as stated above (see Reviewer 2, 2.2). The HNC-FIT Scales were indeed developed and validated for German speaking HNC-patients recruited in Tirol, Austria, only. Although an English translation was provided as part of the original validation study, to date no formal translation and cross-cultural adaption of the instrument has been performed. Thus, both, the generalizability of the observations presented in the study as well as the applicability of the HNC-FIT Scales for languages other than German, are hampered. All authors believe this formal translation and cross-cultural adaptation should be performed urgently.
Finally, the authors agree that the overall results observed in this study are optimistic when compared with other studies, as stated above (see Reviewer 2, 2.3.). While various reasons for this observation have already been included in the Discussion section of the manuscript (line 440-457), additional influential factors (i.e. inclusion and exclusion criteria, categorizing salivary gland and sinonasal cancers as “other site”) were included in the Discussion section.
To emphasize this important issue, various changes were applied to the manuscript. Please refer to Introduction section, line 70-71 and the Discussion section, line 459-476, line 485-489, and line 493-504.
2.8. “The discussion fails to raise the interest of the reader on what could be the practical benefits for the patients and the doctors of adopting the HNC FIT scale in their practice.”
The authors strongly agree on this issue. The HNC-FIT Scales were designed to aid both, patients and physicians. It allows for quick capture and clear presentation of key functional results, filling a gap in HNC outcome assessment. If relevant functional impairments are identified by this simple screening tool, various, more detailed single function assessment tools may supplement the HNC FIT-scales-. Thus, an adequate and timely functional rehabilitation may be initiated.
To raise the interest of the reader, as suggested, an additional paragraph was added to the Conclusion section of the manuscript. Please see the Conclusion section, line 514-519.
2.9. “Reference 5 and 61 are not complete.”
The authors apologize for this imprecision. The mistake was corrected as suggested. Please see the Reference section, line 553 and line 701.

Reviewer 3 Report
In this manuscript, Riechelmann and colleagues reported on the functional outcomes analysis of HNC patients using the HNC-23 Functional InTegrity (FIT) Scales. It was found that the main factors associated with poor functional outcomes in multivariable analysis were recurrence or persistent disease, poor general health, and higher T stage. The referee thinks this manuscript is well organized and well written. It fits the scope of Cancers. Therefore the referee supports its acceptance with minor revisions noted. For example, the words in Figure 2 are too small to be read and should be revised.
This manuscript adapted a validated tool for the rapid clinical assessment of the functional status of head and neck cancer patients based on observable clinical criteria. The topic is relevant in the field because it deals with the functional analysis of clinical data of cancer patients. I consider this study original as I have not seen the same topic being published previously.
Author Response
Dear editors, dear referees,
the authors very much appreciate the constructive and positive comments of the referees about the manuscript “Functional outcomes in head and neck cancer patients” (ID cancers-1661528). All suggested corrections and changes are marked with track changes in the main manuscript.
As requested per MDPI we provided an additional Graphical Abstract for the manuscript as well as a blank copy of the patient informed consent form.
The authors would like to thank the referees for their time and effort since their suggested changes substantially improved the quality of the manuscript. Thank you very much for reconsidering our manuscript for publication in Cancers.
Referee 3
3.1. “… the referee thinks this manuscript is well organized and well written. It fits the scope of Cancers. Therefore the referee supports its acceptance with minor revisions noted.”
All authors involved in this study and the preparation of the manuscript very much appreciate these positive comments.
3.2. “… the words in Figure 2 are too small to be read and should be revised.”
The authors very apologize for the small size of the words in Figure 2. As suggested, the size of the words was increased to improve their readability. Consequently, the figure was turned sideways. Please refer to the Results section of the manuscript, line 234-239.
3.3. “… the topic is relevant in the field because it deals with the functional analysis of clinical data of cancer patients. I consider this study original as I have not seen the same topic being published previously.”
Again, all authors involved in this study and the preparation of the manuscript are delighted to read such positive comments about the manuscript. Thank you very much.
